# From Egoism to Ecoism: Psychedelics Increase Nature Relatedness in a State-Mediated and Context-Dependent Manner

**DOI:** 10.3390/ijerph16245147

**Published:** 2019-12-16

**Authors:** Hannes Kettner, Sam Gandy, Eline C. H. M. Haijen, Robin L. Carhart-Harris

**Affiliations:** Centre for Psychedelic Research, Department of Brain Sciences, Faculty of Medicine, Imperial College London, London W12 0NN, UK; hannes.kettner17@imperial.ac.uk (H.K.); eline.haijen16@imperial.ac.uk (E.C.H.M.H.); r.carhart-harris@imperial.ac.uk (R.L.C.-H.)

**Keywords:** nature relatedness, mental health, well-being, psychedelics, set and setting, acute effects, ego-dissolution, nature exposure

## Abstract

(1) Background: There appears to be a growing disconnection between humans and their natural environments which has been linked to poor mental health and ecological destruction. Previous research suggests that individual levels of nature relatedness can be increased through the use of classical psychedelic compounds, although a causal link between psychedelic use and nature relatedness has not yet been established. (2) Methods: Using correlations and generalized linear mixed regression modelling, we investigated the association between psychedelic use and nature relatedness in a prospective online study. Individuals planning to use a psychedelic received questionnaires 1 week before (N = 654), plus one day, 2 weeks, 4 weeks, and 2 years after a psychedelic experience. (3) Results: The frequency of lifetime psychedelic use was positively correlated with nature relatedness at baseline. Nature relatedness was significantly increased 2 weeks, 4 weeks and 2 years after the psychedelic experience. This increase was positively correlated with concomitant increases in psychological well-being and was dependent on the extent of ego-dissolution and the perceived influence of natural surroundings during the acute psychedelic state. (4) Conclusions: The here presented evidence for a context- and state-dependent causal effect of psychedelic use on nature relatedness bears relevance for psychedelic treatment models in mental health and, in the face of the current ecological crisis, planetary health.

## 1. Introduction

Humans appear to be becoming ever more disconnected from the natural world, and this sense of disconnection is linked to poor mental health as well as ecological neglect and destruction [1,2,3]. Urbanisation is increasing globally [4] and this may be acting to cut people off from nature [5,6]. This growing disconnection from the natural environment appears to be explained, in part, by the increasing usage of electronic entertainment technology [7,8], and is reflected by a shift in Western cultural products away from nature-based content in media such as books, music, and film since the 1950s [9,10]. Considering the global burden of mental health problems, depression being the leading cause of disability worldwide [11], plus the ongoing mass extinction and environmental destruction our planet’s inhabitants are currently suffering due to human actions, identifying an effective way of enhancing nature connectedness would potentially be of great benefit to human well-being and the environment at large [12,13,14,15,16,17,18]. This becomes especially salient in light of the present lack of effective interventions for reducing people’s environmentally destructive behaviour [19].

### 1.1. Psychedelics

A broad range of mental illnesses, including depression [20,21], post-traumatic stress disorder [22], bipolar personality disorder [23], and eating disorders [24] have been linked to feelings of psychological or social disconnection [25]. Connectedness, in a broad sense, is considered a key mediator of psychological well-being [26,27,28,29], and a factor associated with the recovery of mental health [30]. Psychedelics such as psilocybin have been shown to increase the sense of connectedness, comprising three distinguishable aspects: a connection to the self, others, and the world at large [31,32]. “Classic” or serotonergic psychedelics are a subclass of psychedelic compounds, defined by their common mechanism of action, i.e., full or partial agonism at the serotonin 2A (5-HT2A) receptor. 5-HT2A receptors are highly abundant throughout the neocortex, especially in the high-level association regions such as the posterior cingulate cortex [33,34]. Psychedelics are known for their very low physiological toxicity, being non-addictive, and exerting profound effects on human consciousness. Examples of classical psychedelics include N, N-DMT, LSD, psilocybin, mescaline and compounds of the 2C family. Naturally occurring psychedelics such as psilocybin, mescaline, N, N-DMT and lysergic acid derivatives have been employed for celebration, healing and divination purposes by a number of cultures across the world for centuries, or even millennia [35]. The set (psychological context) and setting (sociocultural context) that frames psychedelic usage is known to be a key determinant of experiential outcomes [36]. We currently find ourselves in the midst of what is sometimes referred to as the ‘psychedelic renaissance’, with scientific research into the properties of these compounds expanding rapidly [37]. This recent development follows decades of restrictive drug laws, rendering any research on these substances virtually impossible, as a reaction to the widespread public usage of psychedelics that emerged in the West during the 1960s. This early heyday of Western psychedelic culture coincided with the rapid growth and expansion of the environmental movement [38,39,40,41], which has led some to argue that psychedelic drug use may have contributed to the impetus of modern ecology movements [39,42]. Reflecting these historico-cultural parallels, several recent studies have observed how psychedelics can catalyse measurable and enduring increases in people’s feelings of nature connectedness, or relatedness [43,44].

### 1.2. Effects of Nature Relatedness on Psychological Wellbeing and Pro-Environmental Behaviour

Nature relatedness can be described as one’s level of self-identification and subjective sense of connectedness with nature. It is analogous to the concept of “biophilia”, which is defined as “the connections that human beings subconsciously seek with the rest of life.” [45]. Nature relatedness has previously been found to be increased through time spent in nature [2,46,47,48] and nature contact-based activities, especially when evoking themes of meaning, compassion, and beauty [49]. An increased acknowledgement of nature has also been implicated in enhancing psychological connectedness in a broader sense, including connectedness to other people, nature, and life in general [50]. There is a substantial body of literature linking nature relatedness to psychological health and well-being, including improved well-being at the state and trait level [26,28,46,51,52,53,54,55,56,57,58,59]. Specifically, nature relatedness has been found to be associated with lower levels of anxiety [57,60], a greater perceived meaning in life [26,55], higher vitality [61], higher psychological functioning [62], greater happiness and positive affect [55,57,63]. Nature relatedness has also been found to be a strong predictor of pro-environmental awareness, attitudes, and behaviour [14,16,17,51,64], outperforming all other variables tested as a single construct [65].

Nature relatedness is distinct from nature exposure or immersion, bringing independent and additive benefits, although both of the latter have a reciprocal, synergistic relationship. That is, ratings of nature relatedness were found to correlate with time spent in nature and engagement with nature-based activities [2,66]. Conversely, contact with and time spent in nature predict higher measures of nature relatedness [46,48,49,51,54,67]. Nature relatedness has also been found to be associated with the perceived restorativeness of natural settings [68] and to mediate the effect of nature exposure on affect, with more positive outcomes of nature exposure observed in those who rate highly in nature relatedness [69]. Following exposure to nature, high nature relatedness has been shown to elicit higher valuations of intrinsic (e.g., personal growth, intimacy, community) as opposed to extrinsic (e.g., money, image, fame) aspirations [70]. Measures of well-being have also found to be partially mediated by the degree of nature relatedness in response to perceiving natural beauty [47].

### 1.3. Health Effects of Nature Exposure

A recent large-scale (N = 19,806) UK population study found that regular contact with nature is associated with good physical health and high subjective well-being, with a minimum time of 120 min per week of nature exposure being optimal for good health [71]. A recent meta-analysis pooling data from 143 studies, totalling 290 million participants from 20 different countries, found that spending time in nature was associated with a number of significant physical health benefits, including reduced risk of type II diabetes, cardiovascular disease, premature death, preterm birth and reductions in stress, high blood pressure and cholesterol [72]. Nature exposure has been found to reduce stress hormone markers, such as cortisol levels, including in urban environments [73]. Improved functioning of the immune system has previously been discussed as a key pathway for the health-promoting effects of nature exposure [74], supported by findings such as the increased activity of natural killer (NK) immune cells, with a three day ‘forest-bathing’ trip leading to increased NK activity for up to 30 days [75]. Nature exposure has also been found to aid in recovery from surgery [76] and to improve mood and memory in sufferers of major depressive disorder [77]. Contact with nature has been determined as a strong predictor of overall psychological well-being [78], with a recent meta-analysis of 32 studies (N = 2365) showing that contact with nature is associated with greater measures of hedonic well-being, including a moderate increase in positive affect, along with a small but significant decrease in negative affect [79].

Nature exposure has furthermore been found to increase directed-attention abilities [80], to increase attentional capacity, positive emotions, and the ability to reflect on a life problem, with the positive effects of nature exposure partially mediated by an increase in nature relatedness [46]. Nature exposure has been associated with decreased anxiety [81], decreased stress [82,83], a decrease in rumination [81], increased vitality [84], psychological restoration [85], and enhanced prosocial orientation and net positive affect [50]. Access to nature and to green spaces acts as a buffer against mental health problems, life stress, and adversity in children [86,87] and adults [88] and is associated with significantly lower rates of depression, anxiety, and stress in US neighbourhoods, after controlling for a variety of confounding factors [89]. Enhanced nature exposure through living in greener environments is associated with enhanced physical activity [18], better mental health, and lower all-cause mortality [90,91]. Recent research indicates that even 5 min spent in nature is enough to boost self-esteem and mood [92,93], while a mere half hour spent in nature over the course of a week can reduce depression and lower blood pressure [94]. Nature exposure has also been implicated in elevating value orientation scores and environmental sustainability intentions [95].

### 1.4. Psychedelics and Nature Relatedness

Accounts of feelings of connectedness with nature can be found in a considerable number of anecdotal reports of psychedelic experiences, including early accounts, notably detailed in Masters and Houston [96], who observed people actively seeking out natural environments as settings for their psychedelic experiences. Masters and Houston argued that experiences in natural settings can foster an empathic connection to nature and the humble positioning of one’s self within it, which is less likely to apply to man-made environments. This idea resonates with the frequency of reports of “profound levels of identification or merging with the natural world” [97]) during psychedelic experiences. Based on early therapeutic work, Grof [98] described subjects undergoing LSD therapy sessions reporting a dissolution of boundaries and awe-inducing feelings of unity with nature during peak psychedelic effects. The experience of awe is considered to be a positive experiential component of both contact with nature [99,100,101,102,103] and of psychedelic mystical-type experiences [104], and has previously been linked to enhanced well-being [105], as well as prosociality (intent to benefit others) [99]. Changes in perception of nature following a psychedelic experience can be enduring, with a follow up to Walter Panhke’s ‘Marsh Chapel’ study [106], which employed a single high (30 mg) dose of psilocybin among a cohort of divinity students in a religious setting, reporting that 24–27 years post-experience, a key theme that emerged in the majority of follow-up interviews was an enhanced appreciation for life and nature, among other effects which study participants directly ascribed to their prior psilocybin experience [107].

A recent clinical study evaluating the potential of psilocybin therapy for treatment resistant depression found positive responses in the weeks following treatment, reflected by the large before versus after treatment effect sizes (Cohens d >2) one to five weeks after the main intervention [108]. At a six-month follow-up, six patients still met criteria for remission and 17 of the 20 endorsed that the treatment had helped them in some way, with all 20 reporting a renewed sense of connectedness as a mediating factor [31,32]. This sense of connectedness was commonly experienced acutely during the psilocybin experience, but also appeared to be an enduring change, with participants reporting the effect months following the session. Most typically, participants in the trial reported feeling ‘reminded’ of the value and beauty of nature [32], as the following accounts attest:

“Before I enjoyed nature, now I feel part of it. Before I was looking at it as a thing, like TV or a painting. [But] you’re part of it, there’s no separation or distinction, you *are* it.”(P1)

“I felt like sunshine twinkling through leaves, I *was* nature.”(P8) [32]

### 1.5. Putative Mechanisms of Psychedelic-Induced Changes in Nature Relatedness

It is an increasingly well-established principle that the quality of an individual’s acute experience under a psychedelic is predictive of subsequent long-term psychological outcomes—such as improvements in mental health. For example, experiences of “oceanic boundlessness” were found to be a predictor of a positive response to psilocybin therapy for treatment-resistant depression [109]. The experience of “oneness” or “unity”, a sub-factor of oceanic boundlessness [110], and the loss of self–world boundaries are reliably occasioned by high doses of psychedelics [43,111,112,113] and considered a cornerstone of the mystical-type or peak experience that psychedelics, such as psilocybin, are known to catalyse [35,106,109,114,115,116,117,118,119,120]. Notably, nature relatedness has, in some cases, also been discussed as an experiential sense of oneness with the natural world [51].

In one of the few existing empirical studies on the relationship between psychedelics and nature relatedness in healthy individuals, Nour and colleagues [121] reported that lifetime psychedelic use (but not use of the other substances examined) was predictive of higher ratings on the personality traits of openness, liberal political views, and also nature relatedness. These three factors were also found to be predicted positively by experiences of ego-dissolution during people’s most intense psychedelic experience, pointing towards a potential mediational role for the acute psychedelic experience—not only for clinical responses, but also for changes in nature relatedness [121]. Closely related to the above-mentioned concepts of unity and connectedness, the experience of ego-dissolution has previously been described as “a disruption of ego-boundaries, which results in a blurring of the distinction between self-representation and object-representation” [122] and elsewhere as a “complete loss of subjective self-identity” [123]. Other relevant studies have found that the self-reported values and beliefs of users of classical psychedelics reflected a higher concern for the environment than in both non-users and users of other substances [43,124,125].

Ayahuasca users have been found to rate more highly in self-transcendence [126], which is a significant positive predictor for nature relatedness and environmental concern [48]. In a retrospective survey of 150 psychedelic users, all reported an increase in nature connection following their psychedelic experiences, with 66% also stating that their environmental concern had increased. Changes in attitude also resulted in (self-reported) positive changes in ecologically conscious behaviour among more than half of the sample, with 16% of people even going as far as to change careers following their psychedelic use to what was considered to be more environmentally orientated or ecologically conscious forms of work. Of all the psychedelics examined, psilocybin mushrooms were the most commonly used and associated with increased nature connection and environmental concern [127]. Similarly, one analysis pooled data from eight double-blind placebo-controlled studies where psilocybin was administered to healthy volunteers one to four times and found that 38% of participants retrospectively reported enduring positive changes in their relationship to the environment 8–16 months post-experience [125].

What remains less clear, however, is whether the association between psychedelic use and nature relatedness is merely correlative or causative, i.e., preceding the psychedelic experience, or resulting from it. To our knowledge, longitudinal changes in psychedelic users’ relationships with nature have so far only been reported once in the existing literature; arguably the strongest evidence for a causative influence of psychedelic use on nature relatedness comes from a recent appendage to a clinical trial with psilocybin therapy for treatment-resistant depression [44]. This controlled study measured nature relatedness before and after two sessions with psilocybin for treatment-resistant depression. Nature relatedness scores were increased in the weeks after treatment and remained significantly elevated at a 7–12 months follow-up time-point [44]. The major limitation of this particular study, however, was the absence of a control condition and the very small sample size (N = 8).

The present study aimed to extend on this work and better understand the relationship between nature relatedness and psychedelic use in healthy individuals by collecting a much larger sample and including an assessment of the influence of natural settings and the subjective quality of the psychedelic experience on changes in nature relatedness. This was achieved by accommodating an online survey system on a purpose-built website (www.psychedelicsurvey.com), which collected data on multiple time points before and after a psychedelic experience from individuals that were planning to take a psychedelic through their own initiative.

A positive relationship between nature relatedness and psychedelic use was hypothesised to be reflected by: (1) higher baseline nature relatedness in participants with more lifetime psychedelic use and (2) an increase in nature relatedness two and four weeks after, compared to before the psychedelic experience, which was expected to be correlated with a simultaneous increase in psychological well-being. Furthermore, in line with the previous literature [121], we expected to find the experience of ego-dissolution to be predictive of post-psychedelic increases in nature relatedness. As secondary predictors, the same was expected for mystical-type, but not challenging experiences or visual effects. Lastly, we also tested the hypothesis that the perceived influence of natural settings on the quality of the overall experience would be related to greater increases in nature relatedness, providing the first empirical evidence for a positive effect of natural settings on psychological outcomes following psychedelic use.

## 2. Materials and Methods

### 2.1. Design

The present data were collected between March and November 2017 over the course of a large-scale prospective online survey. Only those measures relevant to the current study will be discussed here; for a full overview of the design see [128].

Through online advertisements on drug-related websites, individuals with a good comprehension of the English language who were planning to use psilocybin/magic mushrooms/truffles, LSD/1P-LSD, ayahuasca, DMT/5-MeO-DMT, *Salvia divinorum*, mescaline, or iboga/ibogaine in the near future were invited to sign up for the study on the purpose-built website www.psychedelicsurvey.com. It was clearly stated on the sign-up page that the coordinators of this study did not endorse the use of psychedelics and no financial or other compensation was offered to participants. Automatic email reminders were then sent out at multiple time-points before and after the indicated date of the experience to participants who signalled consent, including links to surveys hosted on the online platform surveygizmo. The study received a favourable opinion from the Imperial College Research Ethics Committee and was sponsored by the Imperial Joint Research and Compliance Office.

### 2.2. Measures

Measures taken at baseline one week prior to the psychedelic experience that were relevant to the present study included demographics, psychological well-being via the Warwick–Edinburgh Mental Wellbeing Scale [129], the short form of the nature relatedness scale [130], and amount of lifetime psychedelic drug use, using categorical response options ranging from Never to >100 times [121].

One day after the psychedelic experience, measures pertaining to acute subjective drug effects and setting variables were collected. These included the mystical experience questionnaire [131], which assesses positive mood, perceived transcendence of time and space, a sense of ineffability, and mystical feelings as key components of mystical-type peak experiences; the ego-dissolution inventory [122], measuring acute disintegration of the sense of self; and the challenging experience questionnaire [132], which includes items pertaining to fear, grief, physical distress, insanity, isolation, death, and paranoia. As a measure of the visual effects (VE), the audio-visual synaesthesia, complex, and elementary imagery subscales from the altered states of consciousness rating scale were included [110]. To assess the setting, a binary response item specified whether or not the experience took place with access to nature, and, if this item was answered affirmatively, an additional item measured to what extent access to nature was perceived to have influenced the overall quality of the experience, rated on a visual analogue scale (VAS) with values ranging from 0 to 100.

Two and four weeks after the experience, WEMWBS and NR-6 were repeated in order to assess persisting effects on well-being and nature relatedness. Additionally, a two-year follow-up questionnaire included the NR-6 and an item asking for the number of additional psychedelic experiences since the previous time-point.

### 2.3. Statistical Analysis

Due to the categorical measurement of lifetime psychedelic use, a two-tailed Spearman correlation was carried out to assess the association between lifetime psychedelic use and nature relatedness at baseline. A secondary partial correlation was conducted to control for potentially confounding effects of age.

To further investigate a potential causal influence of psychedelic use on nature relatedness, we proceeded to assess the longitudinal change in NR-6 scores from before to after the psychedelic experience. Due to strong negative skew in the distribution of NR-6 scores at each time point (skewness ranging from −1.009 to −1.338), we chose confirmatory generalized linear mixed regression models (GLMMs) to assess whether the psychedelic experience had an effect on nature relatedness and which aspects of the acute experience would mediate this effect. Among various other response distributions, GLMMs are able to accommodate skewed data using gamma regression [133], which was employed in the present study using the glmer procedure in R [134]. All reported GLMMs used maximum likelihood estimation, and the Akaike information criterion (AIC) of goodness of fit was applied to choose random effect structures. To test the primary hypothesis that nature relatedness would change after a psychedelic experience, a first GLMM was constructed containing in the fixed part time as repeated effect and NR-6 as dependent variable. A random intercept was included, accounting for overall differences in the NR-6 scores across subjects. Generalized linear hypothesis tests were carried out with post hoc Tukey contrasts to interpret the nature of the effect [135]. To test for long-term effects, this analysis was repeated with the two-year follow-up included. To account for the potential effects of additional experiences, a two-tailed Pearson correlation between the number of additional psychedelic experiences and NR-6 difference scores between the four-week and two-year endpoints was run.

In order to test whether observed changes in nature relatedness were associated with changes in well-being, two-tailed Pearson correlations were calculated between the difference scores of NR-6 and WEMWBS at the two-week and four-week endpoints. For a separate analysis of well-being changes in the investigated sample, please refer to [128].

To investigate predictors of changes in nature relatedness, time and the interactions of time with z-standardized scores of measures of mystical-type experiences (MEQ), ego-dissolution (EDI), challenging experiences (CEQ), visual effects (VE), and perceived influence of access to nature, measured on a 0–100 VAS, were included in the fixed part of a secondary GLMM. In addition to the random intercept, a random slope was introduced to account for subject heterogeneity. A strong correlation was observed between the EDI and MEQ scores (r > 0.8), suggesting collinearity between these factors. Given that, based on previous findings [121], the EDI was our primarily hypothesised predictor, we chose to repeat the analysis without the MEQ and reported results for both models. Additionally, we repeated the final GLMM with a restricted sample of participants showing average or below-average nature relatedness at baseline (NR-6_BL_ ≤ 4.01; N = 291). This step was performed to shift the sample closer to a demographically similar reference population of college students (M [NR-6_BL_ ≤ 4.01] = 3.23, vs. M = 3.00 in [130]) and allow for greater scope of change in the sample, which was otherwise already comparatively high on nature relatedness at baseline, thus incurring a ceiling effect and the related risk of a Type 2 error. To further corroborate and visualise the findings from this restricted sample GLMM, two-tailed Pearson correlations between the predictor variables and NR-6 difference scores were calculated.

A significance threshold of *p* < 0.05 was set and the correlation strength was interpreted according to the guidelines of r = 0.10, r = 0.30, and r = 0.50 for small, medium, and large effect sizes, respectively [136].

All statistical analyses were conducted in R 3.5.3.

## 3. Results

The following sample sizes were collected for survey time points 1, 2, 3, 4, and 5, respectively: N = 654, N = 379, N = 315, N = 212, and N = 64. The demographic information of the 654 participants who completed the baseline survey is reported in Table 1.

### 3.1. Nature Relatedness at Baseline

Mean nature relatedness at baseline was NR-6_BL_ = 4.01 (SD = 0.87). In comparison, the average (mean) nature relatedness scores of college students recruited at a Canadian university ranged between 3.00 and 3.34 [130]. There was a moderate but highly significant positive correlation between lifetime psychedelic use and nature relatedness at baseline (r = 0.306, *p* < 0.0001, Figure 1), reflected by the mean nature relatedness values ranging from M = 3.52 (SD = 1.03) for psychedelic-naïve participants to M = 4.45 (SD = 0.55) in individuals who indicated more than 100 occasions of psychedelic use. The observed correlation was still highly significant after controlling for age in a partial correlation (r = 0.267, *p* < 0.0001).

### 3.2. Changes in Nature Relatedness

A first confirmatory GLMM revealed a significant main effect of time on NR-6 scores (β = 0.009, *p* = 0.014, Figure 2), indicating an increase in nature relatedness post-psychedelic use. Tukey post hoc tests showed that nature relatedness was significantly higher both two weeks (NR-6_2W_ = 4.13, SD = 0.81, z = 2.208, *p* = 0.027) and four weeks (NR-6_4W_ = 4.12, SD = 0.77, z = 2.253, *p* = 0.024) after the experience, when compared with the baseline (NR-6_BL_ = 4.01, SD = 0.87), and did not change between the two endpoints (z = 0.367, *p* = 0.714); thus, confirming the prior hypotheses.

Pearson correlations between the difference scores for nature relatedness and well-being at each of the endpoints revealed significant positive associations between Δ*NR-6* and Δ*WEMWBS*, i.e., nature relatedness and well-being changed concomitantly. In terms of effect size, this association was moderate for the four-week change scores (r = 0.331, *p* < 0.0001, Figure 3A) and weak for the two-week change scores (r = 0.250, *p* < 0.0001, Figure 3B).

### 3.3. Predictors of Nature Relatedness Change

Results from the final GLMM including measures of ego-dissolution (EDI), challenging experiences (CEQ), visual effects (VE), and perceived influence of nature as predictors of nature relatedness (NR-6) change are reported in Table 2, as well as those of a model that additionally included the MEQ, assessing mystical-type experiences. Across the entire sample (N in analysis = 233), only ego-dissolution significantly predicted changes in nature relatedness, reflected by a significant interaction between time and EDI scores (β = 0.018, *p* = 0.014). This interaction was not significant in the model including the MEQ, which can be explained by the high collinearity between these two variables (r > 0.8). After restricting the analysed sample to individuals with baseline NR-6 scores below the mean of 4.01 (*N* in analysis = 291), the predictive significance of the EDI was reduced to trend level (*p* = 0.072), although the greater potential for change in this sample effectively increased the strength of the association (β = 0.030). In this model, the perceived influence of access to nature had the strongest influence on nature-relatedness changes across all covariates (β = 0.032), although statistical significance here, too, only reached trend level (β = 0.032, *p* = 0.058).

Pearson correlations between the predictors and the change scores Δ*NR-6_BL-2W_* were significant for two of the five predictor variables, namely ego-dissolution (EDI) and influence of nature, but not mystical-type (MEQ), challenging (CEQ), or visual experiences (VE), reflecting the results of the restricted sample GLMM. The scalar item “To what extent did access to nature influence the quality of your overall experience’ showed a moderate association in those that reported access to nature (r = 0.361, *p* = 0.007, Figure 4), whereas the experience of ego-dissolution was only weakly associated with increases in nature relatedness (r = 0.203, *p* = 0.038).

### 3.4. Long-Term Changes in Nature Relatedness

After the collection of a follow-up assessment approximately two years after the experience, the confirmatory GLMM for assessing the effects of time on nature relatedness was repeated including this fourth time point. Similar to the results presented above, time was a highly significant predictor of NR-6 scores (β = 0.013, *p* < 0.0001, Figure 5), indicating an increase in nature relatedness post-psychedelic use. Tukey post hoc tests showed that nature relatedness was increased two weeks (NR-6_2W_ = 4.13, SD = 0.81, z = 1.997, *p* = 0.046) and, although here only at trend level, four weeks (NR-6_4W_ = 4.28, SD = 0.77, z = 1.946, *p* = 0.052) after the experience, when compared with baseline (NR-6_BL_ = 4.01, SD = 0.87). The mean NR-6 scores were highest two years after the experience, reflected by the significantly elevated scores compared with all other timepoints (NR-6_2Y_ = 4.38, SD = 0.62; z = 4.198, *p* < 0.0001, z= 3.080, *p* = 0.002, z = 2.815, p = 0.005, compared with, NR-6_BL_, NR-6_2W_, NR-6_4W_, respectively). Of note, 47 out of the 64 participants (73.4%) who completed the follow-up assessment reported having had at least one additional psychedelic experience since the previous time-point (median number of additional experiences = 2). In the limited sample of individuals who completed the two-year follow-up and also the previous time-point four weeks post-psychedelic (N = 35), the change between these two endpoints did not correlate significantly with the number of additional psychedelic experiences in the same timeframe (r = 0.056, *p* = 0.748).

## 4. Discussion

The present study aimed to investigate the relationship between psychedelic use and nature relatedness using online surveys and a prospective cohort design. Our primary hypothesis of increased nature relatedness following a psychedelic experience was confirmed, providing the first empirical evidence for a causative role of psychedelic use in the enhancement of nature relatedness in a large sample of healthy participants. This represents an important advancement on the correlative association observed between amount of lifetime psychedelic use and nature relatedness in previous studies [43,121] and evidence of a causal increase in nature relatedness in a small sample of patients with treatment-resistant depression treated with psilocybin therapy [44]. Furthermore, it was found, for the first time, that ego-dissolution and natural settings are important mediators of increases in nature relatedness following psychedelic use.

### 4.1. Nature Relatedness and Lifetime Psychedelic Use

It is notable that nature relatedness in the present sample was substantially higher (13–20%) than in demographically similar populations reported by Nisbet and Zelenski [130], even prior to psychedelic use. Given the observed positive relationship between lifetime psychedelic use and nature relatedness [43,121], part of this difference may be explained by the psychedelic-experienced nature of the current sample—implying that prior psychedelic use had already caused an increase in nature relatedness. On the other hand, baseline nature relatedness of the psychedelic-naïve participants in the current sample was still slightly higher than that of the sample collected by Nisbet and Zelenski (in which psychedelic use was not measured), indicating that the intention to take a psychedelic may also reflect underlying psychological traits associated with higher scores on nature relatedness. Nature relatedness has previously been found to be correlated with the personality trait of openness [66,137,138], which also comprises openness to new experiences. It is therefore not unlikely that people who rate highly on openness are both more nature related and also interested in psychedelic experiences, thus linking the intention of psychedelic use and nature relatedness. Lifetime usage of classical psychedelics is correlated with both increased openness and nature relatedness [43,121]. LSD was found to enhance openness two weeks after dosing, with an increase in global brain entropy in sensory and hierarchically higher networks predicting increases in openness, predictive power here being greatest among those who also reported ego-dissolution during the acute experience [139]. Psilocybin has similarly been found to facilitate prospective, long-term increases in openness, with the mystical-type experience acting as a key mediator in the personality change observed [118]. Nature relatedness is positively associated with personality traits such as extraversion, agreeableness, conscientiousness and openness to experiences, and negatively correlated with neuroticism [2]. It is interesting to note that previous research examining the effects of psilocybin therapy on personality structure in depressed individuals has observed significant increases in both openness and extraversion and trend level increases in conscientiousness, along with significant reductions in neuroticism following psilocybin therapy, which were sustained three months after dosing [140].

### 4.2. Post-Psychedelic Increases in Nature Relatedness and Psychological Wellbeing

Despite the comparatively high values already present at baseline, significant increases in nature relatedness were observed after the psychedelic experience across the entire sample of healthy participants, to values that remained stable between two- and four weeks post-experience. Previous research in depressed patients has found nature relatedness to remain significantly elevated 7–12 months post-psilocybin treatment [44]. In the portion of the current sample that completed the two-year follow-up, these changes were furthermore found to be not only sustained, but rather elevated even further after two years. A possible explanation for this additional increase could be a positive-feedback-like effect, whereby the psychedelic-induced sub-acute enhancement of nature relatedness led individuals to subsequently seek more exposure to nature, which in turn further increased feelings of nature relatedness. Similar self-reinforcing positive feedback loops have been identified in other well-being related behaviours, such as showing gratitude, acts of kindness, or spending money on others, rather than oneself [141,142]. Given that nature relatedness has been found to be higher in older people [58], at least part of this long-term increase might, however, also be explained by an ageing effect. Moreover, one must also not discount the possibility of an attrition-bias effect here, whereby those who returned to the survey after two years were more likely to have experienced further increases in nature-relatedness than those who did not. It is also conceivable that subsequent psychedelic experiences may have contributed to the long-term effects on nature relatedness, although the observed absence of a correlation between the frequency of additional psychedelic uses and nature-relatedness changes between four weeks and two years supports the hypothesis that other lifestyle changes would rather have elicited this long-term increase.

A significant positive association was observed between changes in nature relatedness and changes in psychological well-being, which we recently reported to be enhanced post-psychedelic in the same sample [128]. Nature relatedness has been found to be positively correlated with well-being in numerous studies both at the state and trait level [26,28,46,51,52,53,54,55,56,57,58,59]. Such findings are supported by the contemporary evolutionary psychological perspective on the human–nature relationship, as well as the *biophilia* hypothesis [143]; given that our species has spent almost its entire existence in natural environments, by extension we likely have an innate preference for them [144]. In this sense, modern research is demonstrating psychedelics can act as biophilia enhancing agents.

### 4.3. Ego-Dissolution Mediates Psychedelic-Induced Increases in Nature Relatedness

In line with earlier findings [43,121], a positive correlation between nature relatedness and the lifetime use of classical psychedelics was found in the current study. Nour and colleagues [121] additionally observed a positive association between nature relatedness and the extent of ego-dissolution experienced during participants’ “most intense” past psychedelic experience, leading them to suggest that psychedelic-induced ego-dissolution may play a mediational role for changes in nature relatedness. The current finding of ego-dissolution prospectively predicting longitudinal changes in nature relatedness post-psychedelic lends empirical support to the hypothesis that this relationship is causative, rather than merely correlative in nature. With the loss of self-referential boundaries being a defining characteristic of ego-dissolution experiences under psychedelics [123], as well as experiences of awe in nature [99,101], it may be that the loss of perceived boundaries between the self and the other may in turn facilitate an expanded perception of self/nature continuity or overlap, reflected by increased feelings of nature relatedness. This may in turn partly explain the here reported predictive value of perceived influence of natural surroundings during the psychedelic experience for degree of nature relatedness change.

The here observed strong correlation between the experience of ego-dissolution (associated with the loss of self-referential boundaries) and the mystical-type experience is in line with previous findings [122]. Awe being a central aspect of both mystical-type experiences [104] and experiences of nature [99,100,101,102,103], it may be that the experience of awe is an important mediator of the beneficial after-effects reported in association with the latter. A fundamental component of the experience of awe appears to be the experience of “stimuli that are vast, that transcend current frames of reference, and that require new schemata to accommodate what is being perceived” with the identification of one as a “small self” in relation to something larger than oneself [99]. The experience of awe has been found to be conducive to both well-being [105] and prosociality [99]. Interestingly, experiences of awe in nature have been linked to perception of fractal patterns, such as those found in trees, clouds, rain and birdsong [145,146]—which may also hint at the underlying brain mechanisms [34]. It is noteworthy that, as hypothesised in [147], psychedelics have been observed to increase the fractal character of activity in the brain, with visual imagery elicited by psychedelics commonly described as fractal in character [148]. Awe is also considered an important component and mediator of the so-called “overview effect”. The overview effect [149], experienced by many astronauts viewing the Earth from space, has been described as “truly transformative experiences involving senses of wonder and awe, unity with nature, transcendence and universal brotherhood” [29,150]). Experiences of wonder, transcendence, prosociality, and unity or connectedness with nature are commonly associated with both the overview effect and psychedelic mystical-type experiences [98,115,116,117,120,126,149,151,152].

### 4.4. Access to Natural Surroundings during the Psychedelic Experience Enhances Nature Relatedness

Sharing the recent interest in the empirical study of extrapharmacological ‘set and setting’ factors in psychedelic experiences [128,153], we decided to also investigate the influence of the natural settings present during the psychedelic experience on prospective changes in nature relatedness. We found that, at least in the portion of participants with below-average nature relatedness at baseline (compared with the current sample), the perceived influence of access to nature was positively associated with changes in nature relatedness. Of note, baseline nature relatedness in this restricted sample was similar to that in healthy college populations reported previously [130], suggesting that it might actually be more representative of the general population than the total sample of psychedelic users investigated here. A recent large-scale (N = 19,806) UK population study has found that regularly spending time in nature is associated with good health and high subjective well-being [71]. Arguably, by meaningfully connecting with nature during a psychedelic experience (especially so if the experience is within the context of pleasing natural surroundings), otherwise healthy individuals may be enticed to spend more time in nature in the future, thereby adopting healthier, more nature-related lifestyles. It will be interesting to include measures assessing such behavioural changes in future prospective studies on the outcomes of psychedelic use.

It is important to note that only one study has been conducted previously providing controlled, quantitative and prospective data on a psychedelic’s capacity to increase nature relatedness in the long term [44]. The therapeutic setting for the relevant psilocybin sessions was very supportive, with low lighting, carefully selected music and empathic guides present. There were also flowers in the room, but aside from these features, the environment was otherwise far from being nature-enriched. Of note, in spite of this soothing but clinical environment, a long-term increase in nature-relatedness was observed [44]. This is perhaps even more surprising in the context of the findings of Studerus and colleagues [125], who collated data from psilocybin sessions conducted on participants in presumably even more nature-deprived settings, including a PET scanner, indicating that enduring increases in one’s sense of connection to or appreciation of nature are not entirely contingent on the setting in which the relevant psychedelic experience takes place.

### 4.5. Psychedelic-Assisted Nature Exposure for Fostering Greater Environmental Awareness

At the present time, psychedelics are among the most strictly controlled and criminalized substances over much of the world [154]. The context of usage is an essential factor with regard to the beneficial effects of these powerful substances, and neglecting contextual parameters can impose risks to users and lead to a lack of clinical effectiveness [153]. Given the strict regulations in places governing clinical investigations with psychedelics, research is often conducted in medically monitored hospital environments, although care has been taken to incorporate nature imagery into psilocybin therapy rooms. A recent study examining the effects of mindfulness meditation on the psilocybin experience was conducted in the less clinical setting of an alpine meditation retreat centre, where the practice of mindfulness meditation was found to enhance psilocybin’s positive effects while appearing to counteract potential dysphoric or anxiety reactions, with the pristine retreat setting likely contributing to this finding [155]. The current result of natural settings affecting psychedelic-induced nature-relatedness changes, together with the known benefits of increased nature relatedness and exposure to the individual and potentially, society, suggests that providing monitored psychedelic sessions in more natural settings may hold a unique potential and supports the principle of incorporating nature into the psychedelic-therapeutic centres of the future. By fostering nature relatedness, such experiences may be particularly beneficial not only for individual psychological well-being but also at a societal level and for the biosphere beyond - by increasing environmental concern and associated pro-environmental behaviours. A recent online study examining lifetime experiences with classical psychedelics found their usage (but not that of other consumed substances) to strongly predict self-reported pro-environmental behaviours. This relationship was found to be mediated by people’s degree of nature relatedness, particularly their self-identification with nature [43]. Environmental values and behaviours—plus nature relatedness—have been identified as being strongly linked in a number of studies [2,14,16,17,55,64,156], with the latter being the best predictor of ecologically conscious behaviour [65]. Pro-environmental behaviour has been found to be strongly linked to prosocial behaviour, with an increase in one fostering an increase in the other through their mutually enhancing interrelationship [157]. However, it is possible to conceive how self-identification with nature may on the other hand potentially also increase depression, anxiety or stress as a result of increased environmental concern and awareness of local and global ecological degradation [58]. The initially dystonic diminishing of ego-defenses as a result of enhanced sensitivity to the nonhuman world may be seen as an intermediate step to attaining a more meaningful sense of self as part of a wider ecological framework [158]. Considering the intense vulnerability that such states may entail, the importance of a safe setting and psychological support during psychedelic experiences cannot be stressed enough, especially in the presently discussed context of recreational and self-medicative use outside of controlled clinical and laboratory environments [123].

### 4.6. Study Limitations and Future Research

As noted by Haijen and colleagues [128], the web-based observational nature of the here described online survey has several limitations. Most significantly, the recruitment criterion of intent to take a psychedelic substance led to a positive sample bias towards psychedelic use and likely also greater openness towards new experiences in general [128]. In addition, the sample was highly educated and predominantly male, further impairing the generalisability of the present findings. As discussed above, attrition bias may have skewed the sample at later time points into a more positive direction, especially so for the two-year follow-up. The lack of experimental control (e.g., verification of drug dose and purity) is a particularly well recognised limitation of observational studies and, more specifically relevant to this particular paper, meant we were unable to verify the types of environments in which participants took the psychedelic. For this reason, we chose to instead analyse the *perceived influence* of natural surroundings as a predictor of nature relatedness change, but future research on contextual factors influencing psychedelic use might also attempt to gain more fine-grained assessments of the actual surroundings. In a similar vein, future studies investigating the effects of natural environments on psychedelic experiences may benefit from more detailed, specialised, and ideally validated subjective experience measures, rather than using a single-item measure of perceived influence of natural surroundings as was done here. Nevertheless, the presented results make a strong case for the potential value of controlled studies with psychedelics in natural environments. Acknowledging the practical and ethical difficulties in setting up such a study, we propose the use of specific techniques for nature connection before or after treatment with psychedelics, such as forest walking, or *Shinrin-Yoku* (forest bathing), a Japanese form of nature therapy (for recent reviews see [159,160]), as well as the inclusion of natural scenery and artefacts (e.g., flowers) in the treatment space itself. The influence of such factors could be assessed in a classic 2:2 design [153]. Future studies on the relationship between psychedelic use and nature relatedness would benefit from the inclusion of both introspective attitudinal as well as behavioural measures of environmental concern, in addition to related concepts such as consumerism and lifestyle choices, in order to enhance the validity of the findings and avoid common methods bias [161]. To furthermore avoid ceiling issues, inclusion of psychometrically superior instruments such as the Disposition to Connect with Nature scale [162] should be considered, which, for the sake of parsimony, were not selected for the current study.

### 4.7. Societal and Ecological Relevance

It is widely accepted that we have entered a sixth mass extinction event following human actions on the biosphere [163,164,165,166,167]. Reconnecting humans with nature and healing the apparently growing sense of alienation from it should be considered a common and urgent priority area for humanity—and while individual perspective and action holds value, particular responsibility to ensure ecological justice must fall on those who wield the greatest power, namely policy makers and other individuals and organisations who possess significant influence [12,15,168]. It is worth noting here that there presently exists a dearth of effective interventions for reducing people’s environmentally destructive behaviour [169]. It can be considered that “without the direct experience of nature needed to form an ecological consciousness, we cannot expect an ecological conscience which motivates care and action.” [123,170]). The identification of means through which humans can better identify and connect with nature, thereby fostering awareness and acceptance of their “ecological self”, should thus be considered of utmost importance, both for individual well-being and our planet’s future [12,13,14,15,19,43]. In the words of a group of ecopsychologists concluding the ‘Psychology as if the Whole Earth Mattered’ conference almost 30 years ago: “if the self is expanded to include the natural world, behaviour leading to destruction of the world will be experienced as self-destruction.” [12]. As a man-made crisis, climate change is not merely a political—but also a deeply psychological—subject. The loss of connection with nature can be seen as both cause and effect of what the social scientist Renee Lertzman has recently called ‘environmental melancholia’, the self-destructive dissociative state of simultaneous awareness of ecological threat and apathy in the face of it [171]. Given the demonstrated capacity of psychedelics to oppose this pervasive environmental melancholia by enhancing human-nature relatedness, it would seem their widespread prohibition is not in the best interests of our species, or the biosphere at large.

## 5. Conclusions

This paper reports on findings that, when combined with previous work, imply a reliable and robust positive association between psychedelic use and nature relatedness. More specifically, in this particular study, we found a strong relationship between the amount of lifetime use of psychedelics and nature relatedness, as well as increases in nature relatedness from before to after psychedelic use, assessed in a prospective way. The observed increase in nature relatedness post-psychedelic use was correlated with concomitant increases in psychological wellbeing and remained significantly elevated two years after the psychedelic experience. Additionally, the study found that the acute subjective experience of ego-dissolution and the environmental context of the psychedelic experience—i.e., whether it occurred in a natural setting, positively predicted increases in nature relatedness post-psychedelic. Together, these findings point to the potential of psychedelics to induce enduring positive changes in the way humans relate to their natural environments.

## Figures and Tables

**Figure 1 ijerph-16-05147-f001:**
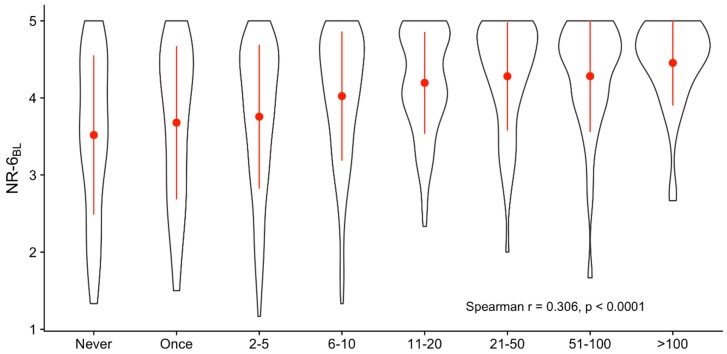
Distribution of nature relatedness depending on frequency of lifetime psychedelic use. A positive correlation was found between nature relatedness at baseline (NR-6_BL_) and lifetime psychedelic use. Red dots and lines represent mean ± SD.

**Figure 2 ijerph-16-05147-f002:**
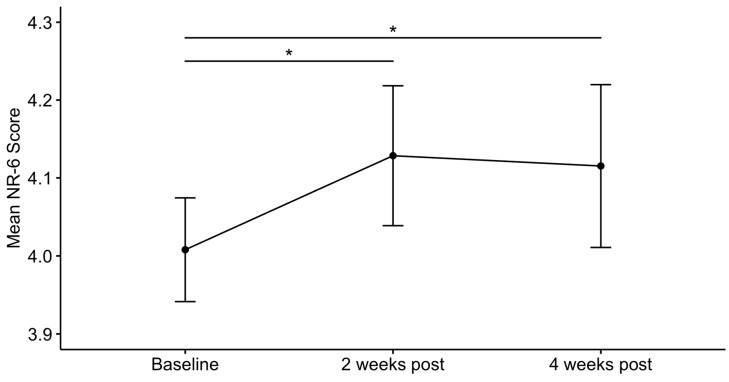
Changes in nature relatedness post-psychedelic. Nature relatedness (NR-6) scores at baseline, i.e., one week before the psychedelic experience, and at two and four weeks after the experience. Error bars represent mean ± 95% CI. * *p* < 0.05.

**Figure 3 ijerph-16-05147-f003:**
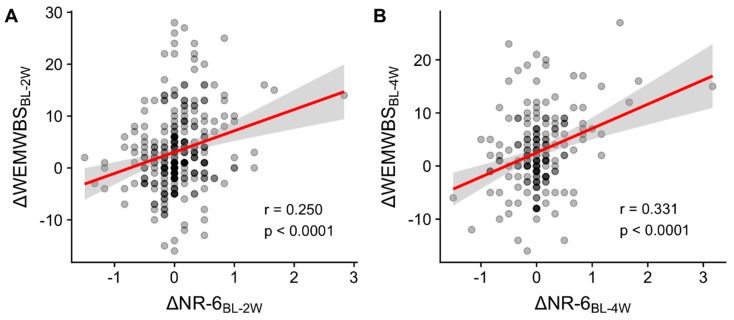
Correlations between changes in well-being and nature relatedness. Increases in nature relatedness (NR-6) were positively associated with increases in well-being (WEMWBS) both two weeks (2W), (**A**) and four weeks (4W), (**B**) after a psychedelic experience.

**Figure 4 ijerph-16-05147-f004:**
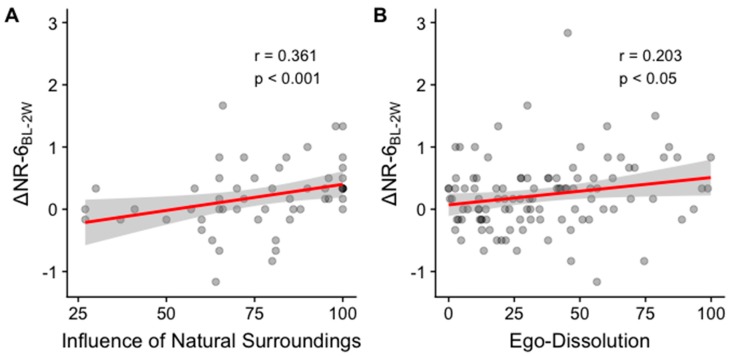
Nature relatedness (NR-6) changes from before to after a psychedelic experience are predicted by acute state and setting. Increase in nature relatedness two weeks (2W) post-psychedelic vs. (**A**) perceived influence of natural surroundings during the experience and (**B**) extent of ego-dissolution during the experience.

**Figure 5 ijerph-16-05147-f005:**
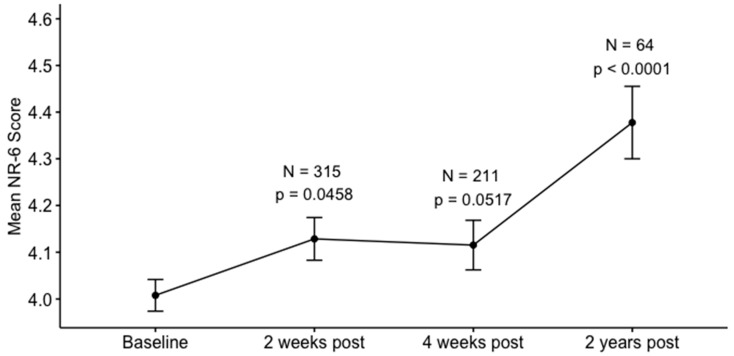
Long-term changes in nature relatedness. Nature relatedness (NR-6) scores are displayed at baseline, i.e., one week before the psychedelic experience, at two weeks, four weeks, and two years after the experience. P values indicate significant differences between the endpoints and baseline derived from Tukey contrasts. Error bars represent mean ± 95% CI.

**Table 1 ijerph-16-05147-t001:** Demographic data collected in the first survey.

**Total**		654
**Gender**	Male	485 (74.2%)
	Female	165 (25.2%)
	Other	4 (0.6%)
**Age**		28.9 ± 10.4
**Educational Level**	Left school before age 16 without qualifications	8 (1.2%)
	Some high school/GCSE level (in UK)	45 (6.9%)
	High school diploma/A-level education (in UK)	97 (14.8%)
	Some university (or equivalent)	179 (27.4%)
	Bachelor’s degree (or equivalent)	193 (29.5%)
	Post-graduate degree (e.g., Masters or Doctorate)	132 (20.2%)
**Employment Status**	Student	256 (39.1%)
	Unemployed	53 (8.1%)
	Part-time job	98 (15.0%)
	Full-time job	237 (36.2%)
	Retired	10 (1.5%)
**Nationality**	United States	199 (30.4%)
	United Kingdom	128 (19.6%)
	Denmark	60 (9.2%)
	Germany	32 (4.9%)
	Canada	32 (4.9%)
	Other (50 in total)	203 (31.0%)
**Previous Psychedelic Drug Use**	Never (psychedelic naïve)	62 (9.5%)
	Once	40 (6.1%)
	2–5 times	148 (22.6%)
	6–10 times	106 (16.2%)
	11–20 times	109 (16.7%)
	21–50 times	110 (16.8%)
	51–100 times	39 (6.0%)
	More than 100 times	40 (6.1%)

*Note.* Means ± standard deviations and absolute frequencies are shown. Numbers in parentheses indicate the percentages corresponding to the absolute frequencies. See also [128].

**Table 2 ijerph-16-05147-t002:** Predictors of psychedelic-induced changes in nature relatedness.

	N = 698 All Predictors		Excluding MEQ		N = 291 NR-6_BL_ ≤ 4.01	
	Estimate	*p*-Value	Estimate	*p*-Value	Estimate	*p*-Value
**Fixed Effects:**						
Intercept	3.164	-	3.164	-	2.900	-
Time	0.023	0.196	0.017	0.286	0.031	0.425
Time × EDI	0.013	0.175	0.018	0.014 *	0.030	0.072 ^†^
Time × CEQ	−0.0002	0.581	−0.0002	0.709	−0.0004	0.719
Time × VE	<0.0001	0.764	<0.0001	0.972	0.0005	0.485
Time × Nature	0.006	0.355	0.007	0.305	0.032	0.058 ^†^
Time × MEQ	0.009	0.451	-	-	-	-
**Random Effects:**						
σ_int_	0.061	-	0.061	-	0.081	-
σ_slope_	0.008	-	0.007	-	0.013	-
Model Fit (AIC)		2276.8		2275.4		860.0

EDI: Ego-dissolution inventory; CEQ: Challenging experience questionnaire; VE: Visual effects subscale of the 11-dimension altered states of consciousness questionnaire; Nature: Perceived influence of nature on the overall quality of the experience, measured on a 0–100 VAS scale; MEQ: Mystical experience questionnaire; AIC: Akaike information criterion. * *p* <0.05; ^†^
*p* <0.1.

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
