# Peer review of "From Egoism to Ecoism: Psychedelics Increase Nature Relatedness in a State-Mediated and Context-Dependent Manner"

_ijerph, 2019, doi:10.3390/ijerph16245147_

Round 1
Reviewer 1 Report
I congratulate the authors for their well-written paper and their effort in undergoing this extensive longitudinal study. However, as there is always room for improvement, I would like to offer the following comments:
Language
I am not a native speaker, and thus, cannot give reliable advice on the language. However, in my opinion the language of this manuscript is at a native’s speaker level.
Introduction
The introduction is well written, and the basis for the research question of the study is laid out well.
However, structure could be enhanced a bit further by focusing more on one issue in each section and using subheadings. I am missing an introduction into or overview of psychedelic substances and their uses, because this is not common knowledge. This should include a differentiation between psychedelic and less-psychedelic but probably more addictive substances. In addition to a condensed overview on such substances, a description of their common uses and common circumstances under which they are consumed should be included in the manuscript (i.e., settings, peer groups, recreational or not, …).
The authors review nature relatedness and its relation to pro-environmental behavior shortly in the introduction. I think the authors should stress this point a bit more, because they present a way to increase nature-relatedness, which in turn is related to ecological or in a wider sense to sustainable behavior. This will help to lay the ground for their discussion on the same topic in the discussions section.
Method
I was able to follow the detailed description of the methods very well. However, I am wondering why the authors did not include a Figure and analysis for the effect on well-being similar to Figure 2?
Please use comprehensible descriptions/labels for the constructs throughout text and figures (e.g., well-being instead of WEMWBS…) - this will enhance readability substantially. This holds for the other chapters as well.
The authors themselves describe the celling effect of their nature connectedness measure. In their discussion they should shortly discuss other superior measures that have no ceiling issues (e.g., Brügger, Kaiser, & Roczen, 2011).
Discussion
The discussion is well supported by the results. However, just like the introduction, it would benefit from a more visible structure, which could be increased through some subheadings.
The authors discuss openness as a variable that potentially explains the relation between nature relatedness and use of psychedelic substances. To my impression, this is the most promising lead for future research. I am not sure if there is much more literature to support this reasoning, but if so, I would very much like to see this.
Starting at line 460 the authors discuss wider effects of fostering nature relatedness, for instance on pro-environmental behaviors. I welcome this outlook and would like to encourage the authors to extend this to even prosociality. Neaman, Otto, and Vinokur (2018) support such implications.
The study’s measures of well-being and nature connectedness both use cognitive measures, which can lead to a substantial common method bias (Podsakoff, MacKenzie, Lee, & Podsakoff, 2003), which has been shown to apply strongly in research on environmental attitude and behavior (Otto, Kröhne, & Richter, 2018). This should be discussed under limitations.
Brügger, A., Kaiser, F. G., & Roczen, N. (2011). One for all? Connectedness to nature, inclusion of nature, environmental identity, and implicit association with nature. European Psychologist, 16, 324-333.
Neaman, A., Otto, S., & Vinokur, E. (2018). Toward an integrated approach to environmental and prosocial education. Sustainability, 10, 1–11.
Otto, S., Kröhne, U., & Richter, D. (2018). The dominance of introspective measures and what this implies: The example of environmental attitude. PLOS ONE, 13(2), e0192907.
Podsakoff, P. M., MacKenzie, S. B., Lee, J.-Y., & Podsakoff, N. P. (2003). Common method biases in behavioral research: A critical review of the literature and recommended remedies. Journal of Applied Psychology, 88, 879-903.
Author Response
The authors would like to kindly thank reviewer 1 for his positive and constructive comments. We feel that by addressing them, we could genuinely improve the quality of the manuscript. In the following, specific changes to the manuscript are detailed in red.
Introduction
However, structure could be enhanced a bit further by focusing more on one issue in each section and using subheadings. Added subheadings to the introduction
I am missing an introduction into or overview of psychedelic substances and their uses, because this is not common knowledge. This should include a differentiation between psychedelic and less-psychedelic but probably more addictive substances. In addition to a condensed overview on such substances, a description of their common uses and common circumstances under which they are consumed should be included in the manuscript (i.e., settings, peer groups, recreational or not, …). A section 'Psychedelics' was added to the introduction
The authors review nature relatedness and its relation to pro-environmental behavior shortly in the introduction. I think the authors should stress this point a bit more, because they present a way to increase nature-relatedness, which in turn is related to ecological or in a wider sense to sustainable behavior. This will help to lay the ground for their discussion on the same topic in the discussions section. It was added that in several of the mentioned references, not only pro-environmental awareness and attitudes, but also behaviour was found to be associated with nature relatedness
Method
However, I am wondering why the authors did not include a Figure and analysis for the effect on well-being similar to Figure 2? A sentence was added under 2.3.: "For a separate analysis of well-being changes in the investigated sample, please refer to [123]"
Please use comprehensible descriptions/labels for the constructs throughout text and figures (e.g., well-being instead of WEMWBS…) - this will enhance readability substantially. This holds for the other chapters as well. Abbreviations were replaced by and/or complemented with comprehensible / spelled-out labels throughout the manuscript to improve readability. Translations of abbreviations were furthermore added to all figure descriptions
The authors themselves describe the celling effect of their nature connectedness measure. In their discussion they should shortly discuss other superior measures that have no ceiling issues (e.g., Brügger, Kaiser, & Roczen, 2011). Sentence added: "To furthermore avoid ceiling issues, inclusion of psychometrically superior instruments such as the Disposition to Connect with Nature scale [156] should be considered, which, for the sake of parsimony, were not selected for the current study."
Discussion
The discussion is well supported by the results. However, just like the introduction, it would benefit from a more visible structure, which could be increased through some subheadings. Subheadings added to the discussion
The authors discuss openness as a variable that potentially explains the relation between nature relatedness and use of psychedelic substances. To my impression, this is the most promising lead for future research. I am not sure if there is much more literature to support this reasoning, but if so, I would very much like to see this. Following sentences were added: "Lifetime usage of classical psychedelics is correlated with both increased openness and nature relatedness [39, 116]. Experience with LSD was found to increase openness two weeks after dosing, with an increase in global brain entropy in sensory and hierarchically higher networks predicting increases in openness, with the latter’s predictive power greatest among those who reported ego-dissolution during the acute experience [134]. Psilocybin has also been found to facilitate prospective, long-term increases in openness, with the mystical-type experience acting as a key mediator in the personality change observed [113]"
Starting at line 460 the authors discuss wider effects of fostering nature relatedness, for instance on pro-environmental behaviors. I welcome this outlook and would like to encourage the authors to extend this to even prosociality. Neaman, Otto, and Vinokur (2018) support such implications. Sentence added: "Pro-environmental behaviour has been found to be strongly linked to prosocial behavior, with an increase in one fostering an increase in the other through their mutually enhancing interrelationship [151]. "
The study’s measures of well-being and nature connectedness both use cognitive measures, which can lead to a substantial common method bias (Podsakoff, MacKenzie, Lee, & Podsakoff, 2003), which has been shown to apply strongly in research on environmental attitude and behavior (Otto, Kröhne, & Richter, 2018). This should be discussed under limitations. Sentence added: "Future studies on the relationship between psychedelic use and nature relatedness would benefit from inclusion of both introspective attitudinal and behavioural measures of environmental concern and related concepts such as consumerism and lifestyle choices to enhance the validity of the findings and avoid common methods bias [155]."
With kindest regards
Reviewer 2 Report
The authors touch upon a very salient effect produced by the psychedelic drug action, that is a feeling of connectedness with the environment, particularly with the nature. The study was designed as an online survey with several measurement points. It delivers evidence that the psychedelic compounds increase the nature relatedness, in relation to an augmented well-being and an ego-dissolution effect. Among the strengths of this work are a high sample size, reasonable statistical approach, the fact that it’s well written and presents an interesting topic and finding. The study also deals with limitations intrinsic to such types of studies, i.e., a self-selection bias and no specific control group. From my perspective, this work is definitely worth considering for publication, if it is within the journal scope. I would like the authors to address the points listed below.
Abstract
Page 1, line 12: “is linked to poor mental health”: I would suggest using a slightly different wording, like “paralleled by”, “coincides in time with”, or something similar? The message is rather clear, but I would rather avoid the assumption that there is an obvious direct link between the two.
1. Introduction
Page 1, line 36: can the authors state more precisely in which sense the cultural products shifted away from nature? Do you mean the presence of nature-related topics in these media? I would argue this has never been a total shift, if so, then rather a tendency.
Page 2-3, lines 86-99: I would appreciate mentioning somewhere a few biological mechanisms with references why the exposure to nature or spending time in the nature/non-urban areas may bring about health benefits (e.g. cortisol levels, oxygenation, etc..).
Page: 3, line 138: “ego dissolution” is sometimes written “ego-dissolution” throughout the text. If there is no other special reasons, please make it consistent.
Page 3, line 139: please delete “;”
Page 4, line 162: “in the existing literature”. I’m curious whether the authors had a chance to access the studies conducted in the early times of psychedelic research in the fifties and sixties, and found anything relevant for this article? I’m aware of the fact that such sources might not be easily found, however I would be happy to know whether the authors extended their searches to these past studies.
Page 4, line 174: out of curiosity I wanted to look at the web page, but seemingly it’s no longer active.
2. Materials and methods
Page 4-5, Design: was there any financial compensation or any other gratification for participating in the survey?
Page 5, line 201 (Measures): were all these measures in English? As a substantial part of the participants reported different nationalities, did you ask about the native language? Depending on the situation, the language differences should be seen as a limitation.
Page 5, lines 207-211: given the fact, this is an interdisciplinary journal, I would welcome the explanation what these scales are measuring.
Page 5, line 223: what about sex?
Page 5, line 237: what do you mean by the “post-hoc Tukey style”?
Page 6: lines 246-248: Have you considered doing the same analysis with MEQ only? I would welcome some additional elaborations on these two scales in consideration of the fact they correlated with each other on the one hand, on the other hand you specify your hypothesis for EDI.
3. Results
Page 10: was there any sort of control what happened in the 2-way period in terms of possible repeated exposures to psychedelic substances?
4. Discussion
Page 13, lines 453-455: In the very recent article published in Scientific Reports, an experiment involving the administration of psilocybin took place in an aesthetically-pleasing environment of a meditation center, in combination with a 5-day long mindfulness-based training. This would be an interesting example of exception to this standard hospital/medical framework.
Discussion, page 13: I would welcome mentioning that the psychedelic substances that are the object of this work, are for the most scheduled psychoactive drugs in most countries, and should not be discussed without possible risks. A reference to one of few existing works on harm potential or safety recommendations in hallucinogen research (e.g. Johnson et al., Human hallucinogen research: guidelines for safety.) would give the reader a balanced view on this point.
Author Response
The authors would like to express their sincere gratitude for the precise and constructive comments which we feel have helped to significantly improve the quality of the submission. Please find point-by-point responses in red
Abstract
Page 1, line 12: “is linked to poor mental health”: I would suggest using a slightly different wording, like “paralleled by”, “coincides in time with”, or something similar? The message is rather clear, but I would rather avoid the assumption that there is an obvious direct link between the two. Wording changed to : "There appears to be a growing disconnection between humans and their natural environments which has previously been linked to poor mental health and ecological destruction." to stress that this link has been suggested in previous research, rather than being established fact
Introduction
Page 1, line 36: can the authors state more precisely in which sense the cultural products shifted away from nature? Do you mean the presence of nature-related topics in these media? I would argue this has never been a total shift, if so, then rather a tendency.Sentence changed to: "This growing disconnection from the natural environment appears to be linked, in part, to increasing usage of electronic entertainment technology [7], and is reflected by a shift in Western cultural products away from nature based content in media such as books, music, and film since the 1950’s [8, 9]. " Shift here would not have to be understood as total shift, but rather a 'movement' in a certain direction (away from nature), following the terminology in the respective reference by Kesebir & Kesebir (2017): "Studying works of popular culture in English throughout the 20th century and later, we document a cultural shift away from nature, beginning in the 1950s."
Page 2-3, lines 86-99: I would appreciate mentioning somewhere a few biological mechanisms with references why the exposure to nature or spending time in the nature/non-urban areas may bring about health benefits (e.g. cortisol levels, oxygenation, etc..). Sentences added: "Nature exposure has been found to reduce stress hormone markers, such as cortisol levels, including in urban environments [71]. Improved functioning of the immune system has previously been discussed as a key pathway for the health-promoting effects of nature exposure [72], supported by findings such as increased activity of natural killer (NK) immune cells, with a three day 'forest-bathing' trip leading to increased NK activity for up to 30 days [73]"
Page: 3, line 138: “ego dissolution” is sometimes written “ego-dissolution” throughout the text. If there is no other special reasons, please make it consistent. Changed to hyphenated 'ego-dissolution' throughout the document
Page 3, line 139: please delete “;”
Page 4, line 162: “in the existing literature”. I’m curious whether the authors had a chance to access the studies conducted in the early times of psychedelic research in the fifties and sixties, and found anything relevant for this article? I’m aware of the fact that such sources might not be easily found, however I would be happy to know whether the authors extended their searches to these past studies. Sentence added: "Based on early therapeutic work, Grof [96] described subjects undergoing LSD therapy sessions reporting a dissolution of boundaries and awe-inducing feelings of unity with nature during peak psychedelic effects."
Page 4, line 174: out of curiosity I wanted to look at the web page, but seemingly it’s no longer active. Apologies, an outdated link was provided! Section changed to "This was achieved by accommodating an online survey system on a purpose-built website (www.psychedelicsurvey.com)"
2. Materials and methods
Page 4-5, Design: was there any financial compensation or any other gratification for participating in the survey? Added "It was clearly stated on the sign-up page that the coordinators of this study did not endorse the use of psychedelics and no financial or other compensation was offered to participants. "
Page 5, line 201 (Measures): were all these measures in English? As a substantial part of the participants reported different nationalities, did you ask about the native language? Depending on the situation, the language differences should be seen as a limitation. Changed sentence to "Through online advertisements on drug-related websites, individuals with a good comprehension of the English language who were planning to use psilocybin/magic mushrooms/truffles, LSD/1P-LSD, ayahuasca, DMT/5-MeO-DMT, Salvia divinorum, mescaline, or iboga/ibogaine in the near future were invited to sign up for the study on the purpose-built website www.psychedelicsurvey.com."
Page 5, lines 207-211: given the fact, this is an interdisciplinary journal, I would welcome the explanation what these scales are measuring. Passage changed to:"These included the Mystical Experience Questionnaire [MEQ, 129], which assesses positive mood, perceived transcendence of time and space, a sense of ineffability, and mystical feelings as key components of mystical-type peak experiences; the Ego-Dissolution Inventory [EDI, 120], measuring acute disintegration of the sense of self; and the challenging experience questionnaire [CEQ, 130], which includes items pertaining to fear, grief, physical distress, insanity, isolation, death, and paranoia."
Page 5, line 223: what about sex?
-> Even though it is likely that males in the sample would have used more psychedelics, we had no prior reason to expect that sex would have an impact on the relationship between liftetime use and nature relatedness - wheras lifetime usage logically had to be assumed to accumulate with increasing age, which is also known to correlate with nature relatedness, thereby introducing becoming a potential confounder.
Page 5, line 237: what do you mean by the “post-hoc Tukey style”? The indeed curious wording "style" was removed and changed simply to 'post-hoc Tukey contrasts' - Tukey’s method is the most common type of post-hoc test when comparing all possible group pairings.
Page 6: lines 246-248: Have you considered doing the same analysis with MEQ only? I would welcome some additional elaborations on these two scales in consideration of the fact they correlated with each other on the one hand, on the other hand you specify your hypothesis for EDI. This was done on the basis that ego-dissolution, but not mystical-type experiences has been associated with psychedelic-induced nature relatedness previously (in Nour et al. 2017). Sentence changed to: "Given that, based on previous findings [119], the EDI was our primarily hypothesised predictor, we chose to repeat the analysis without the MEQ and reported results for both models. "
3. Results
Page 10: was there any sort of control what happened in the 2-way period in terms of possible repeated exposures to psychedelic substances? Yes, additional exposures are now reported and an additional correlational analysis (n.s.) between frequency of additional exposures and nature relatedness changes between the four-week and two-year endpoint was included in the manuscript. Discussion:
"It is also conceivable that subsequent psychedelic experiences may have contributed to the long-term effects on nature relatedness, although the observed absence of a correlation between the frequency of additional psychedelic uses and nature relatedness changes between four weeks and two years supports the hypothesis that other lifestyle changes would have elicited this long-term increase."
4. Discussion
Page 13, lines 453-455: In the very recent article published in Scientific Reports, an experiment involving the administration of psilocybin took place in an aesthetically-pleasing environment of a meditation center, in combination with a 5-day long mindfulness-based training. This would be an interesting example of exception to this standard hospital/medical framework. Added "A recent study examining the effects of mindfulness meditation on the psilocybin experience was conducted in the less clinical setting of an alpine meditation retreat centre, where the practice of mindfulness meditation was found to enhance psilocybin’s positive effects while appearing to counteract potential dysphoric or anxiety reactions, with the pristine retreat setting likely contributing to this finding [149]."
Discussion, page 13: I would welcome mentioning that the psychedelic substances that are the object of this work, are for the most scheduled psychoactive drugs in most countries, and should not be discussed without possible risks. A reference to one of few existing works on harm potential or safety recommendations in hallucinogen research (e.g. Johnson et al., Human hallucinogen research: guidelines for safety.) would give the reader a balanced view on this point. Added sentences: "At the present time, psychedelics are among the most strictly controlled and criminalized substances over much of the world [151]. The context of usage is an essential factor with regard to the beneficial effects of these powerful substances, and neglecting contextual parameters can impose risks to users and lead to a lack of clinical effectiveness [150]. ", " In face of the intense vulnerability that such states may entail, the importance of a safe setting and psychological support during psychedelic experiences cannot be stressed enough, especially in the presently discussed context of recreational and self-medicative use outside of controlled clinical and laboratory environments [118]."
With the kindest regards
Reviewer 3 Report
Thank you very much for this interesting manuscript. Please find my comments in the attached pdf-document.

Author Response
The authors would like to thank the reviewer for his concise and helpful comments. They enabled us to improve the quality of the manuscript substantially. Point-by-point responses are listed below in red.
General comments
Furthermore, a short discussion on possible negative or critical effects of the interventions would be appreciated. One general example: You write “Nature relatedness has also been found to be a strong predictor of pro-environmental awareness and attitudes…”. Anything about “behavior”? Sentence changed: "Nature relatedness has also been found to be a strong predictor of pro-environmental awareness, attitudes, and behaviour…" - indeed all the here referenced studies also included measures of pro-environmental behaviour.
I wonder about the enhanced subjective meaning (e. g. Preller et al.,2017) and suggestibility (e. g. by your group, Carhart-Harris et al., 2015) induced by psychedelic substances. Couldn’t it be that by feeling connected to nature, participants simply think that they are more nature related, but do not actually act accordingly? E. g.,there are recent discussions on the potentially problematic side-effects of the ayahuascatourism or the effects of the global use of ayahuasca on the local populations of the plants in the amazon. Further studies should investigate the actual long-term behavioral changes towards the environment (nature and fellow human beings). We strongly agree with all of these concerns and are planning to include measures on explicit pro-environmental behaviours in the next generation of psychedelic survey studies which is currently in the making.
Nevertheless, this study contributes essentially to the advancement of this important field of research and – as the authors already depict nicely – will serve as foundation for further studies.
Please see below for more detailed comments and concrete examples.
Introduction
The introduction is well-written and gives a good overview on the existing literature both on psychedelics as well as nature-relatedness and their intersection. The relevance becomes clear. I miss some concepts you do not introduce but write about in the discussion (e. g. the experience of “awe”, “prosociality” and especially the term “biophilia”). The reader could profit from an early introduction of the concepts you include in the discussion. Additionally, subtitles could give some structure. Subheadings were added to both introduction and discussion. Definitions of the mentioned concepts were added to the introduction.
Methods
You acknowledged most of my methodological concerns in your limitation section, for which I give you great credit. Here some points:
- Selection bias:
o People that seek psychedelic experiences might already have gone through some processes towards more connectedness and nature relatedness. This might already be a purpose of these people and their attitude might already be towards caring for nature. Indeed! The selection bias is well reflected by the unusually high baseline nature relatedness of the sample - we hoped to counteract this issue by cutting down the analysed sample to those below the mean, thereby approximating the average of reference populations found in earlier research with the same instrument (see under 2.3.:"Additionally, we repeated the final GLMM with a restricted sample of participants showing average or below-average nature relatedness at baseline (NR-6BL ≤ 4.01; N = 291). This step was performed to shift the sample closer to a demographically similar reference population of college students (M[NR-6BL ≤ 4.01] = 3.23, vs M = 3.00 in [128]) and allow for greater scope of change in the sample, which was otherwise already comparatively high on nature relatedness at baseline – thus, incurring a ceiling-effect and related risk of Type 2 error.")
o According to the demographics table, the mean educational level is above general population average. Added sentence: "Furthermore, the sample was highly educated and predominantly male, further impairing the generalisability of the present findings. "
- Systematic drop-out: This is quite a big dropout rate… It could be that especially the participants that did not have a somehow special / intense experience (in any direction?) were those who did not respond to the 2-years-follow-up. Sentence added: "As discussed above, attrition bias may have skewed the sample at later time points into a more positive direction, especially for the two year follow-up."
- What are the underlying mechanisms/mediators? Personality traits such as absorption might be the “cause” for “deeper” experiences (state-dependent) and also for generally increased nature-relatedness. Indeed this is what we found in the present sample, as reported in Haijen et al., 2018: https://www.frontiersin.org/articles/10.3389/fphar.2018.00897/full
- And what about the “frequency of lifetime psychedelic use and its correlation with nature relatedness at baseline? Does that still increase after that specific experience? If you say so, how would you exclude other types of experiences made in between? Did you check/ask how long ago the last psychedelic experience was? Why would the values increase after a second, third, … experience? And also, people for which this was the very first psychedelic experience might have kept on taking these substances which could potentially contribute to the follow-up measures. Generally: What do the people do between the psychedelic experience they refer to and the follow up? This is an excellent point that led us to include an additional descriptive report of subsequent psychedelic experiences, as well as a correlation between frequency of subsequent experiences and changes in nature relatedness between 4 weeks and 2 years. (n.s.)
- Why did you not include the whole 11D-ASC, but only certain subscales? Only the visual subscales were included so as to avoid potential collinearity with the other included measures MEQ, CEQ, and EDI.
Results
- Only minor issues. Please see section E.
Discussion
- Subtitles would be helpful. Subheadings were added to the discussion
- Conceptual understanding:
o What exactly is a natural setting / access to nature? Did you ask about the specific characteristics of the settings? Collecting data on the setting in general (broader concept, beyond nature-related or not) could potentially yield insights in its importance beyond the nature-related aspects.Unfortunately, no more specific information was collected on the details of the natural environment: "[...] future research on contextual factors influencing psychedelic use might also attempt to gain a more fine-grained picture of the actual surroundings. In a similar vein, future studies investigating the effects of natural environments on psychedelic experiences may benefit from more detailed, specialised, and ideally validated subjective experience measures, rather than using a single-item measure of perceived influence of natural surroundings as was done here"
o What exactly would be the implication of increased nature relatedness for the development of treatment models? How would that treatment look like (except for more access to nature)? "Acknowledging the practical and ethical difficulties in setting up such a study, we propose the use of specific techniques for nature connection before or after treatment with psychedelics, such as forest walking, or Shinrin-Yoku (forest bathing), a Japanese form of nature therapy (for recent reviews see [156, 157]), as well as the inclusion of natural scenery and artefacts (e.g. flowers) in the treatment space itself. The influence of such factors could be assessed in a classic 2:2 design [150]."
- You’re talking about the importance of the context in terms of its access to nature or perceived influence of nature. I wonder about other characteristics of 1) the context and 2) the substance that might correlate with that specific aspect (nature) and that might also contribute to the change in nature-relatedness. Couldn’t it be, that for example ayahuasca is more likely to be taken in a group setting, facilitated by nature related and very careful people and in a natural and safe setting, all aspects potentially contributing to the increased nature-relatedness? I would expect that especially the human factor (facilitator or people around) would influence the experience significantly, as it’s well known for example from psychotherapy research, that the psychotherapeutic relationship (a non-specific factor) is one of the strongest predictors of psychotherapeutic outcome. These are excellent points that we are hoping to address with our ongoing 'ceremony survey' which focusses on social dimensions of the psychedelic experience, specifically in retreat settings: www.ceremonystudy.com In the currently investigated sample, the number of ayahuasca users was unfortunately so low that a stratification by substance/setting would not have been meaningful.
- “… personality structure in depressed individuals has observed significant increases in both openness and extraversion and trend level increases in conscientiousness along with significant reductions in neuroticism following psilocybin therapy.” Personality traits are by definition long-term (mean absolute values naturally change across the lifespan within the general population). Are the above-mentioned effects long-term?
Changed to: "It is interesting to note that previous research examining the effects of psilocybin therapy on personality structure in depressed individuals has observed significant increases in both openness and extraversion and trend level increases in conscientiousness along with significant reductions in neuroticism following psilocybin therapy that were sustained three months after dosing [138]."
- “…, at least part of this long-term increase might however also be explained by an ageing effect.” After only two years? I think, here you are too critical about your own findings. This might have indeed been too self-critical. However, given the relatively young sample (average age = 29 years), an age-group where presumably much lifestyle-based nature connection (e.g. through travelling) occurs, we did not want to discard this possibility.
- “Moreover, one must also not discount the possibility of an attrition-bias effect here, whereby those who returned surveys at 2 years were more likely to have experienced further increases in nature-relatedness than those who did not.” It cannot be excluded, that they had had further psychedelic experiences and the last one might not have been 2 years ago. Indeed most participants had further psychedelic experiences. This is now reported and discussed in the manuscript.
Some examples of where/how the writing could be more concise or clearer or something could be added.
Introduction:
- You repeat “connection to self, others, and the world at large”, could be changed Repetition removed
- “… with 16% of people even going as far to change careers…”. What’s the proenvironmental thing about that? Changed to "...with 16% of people even going as far to change careers following their psychedelic use to what was considered to be more environmentally orientated or ecologically conscious forms of work"
- “Of all the psychedelics examined, increased nature connection and environmental concern was most commonly reported in association with the use of psilocybin mushrooms.” Relative or absolute? Changed to "Of all the psychedelics examined, psilocybin mushrooms were the most commonly used and associated with increased nature connection and environmental concern "
- “To our knowledge, longitudinal changes in psychedelic users’ relationship with nature have so far only been reported once in the existing literature.” Source? reference added
Methods:
- “… rated on a scale from 0 to 100.” Visual analogue scale (VAS)? changed
Results:
- “…the perceived influence of access to nature was also predictive of changes in NR-6 scores, however only at a trend level)”. If you define a level of significance (here a=0.05), then a not-significant result should not be described as “predictive”. (There are more examples in the results section.) Wording changed accordingly: "In this model, the perceived influence of access to nature had the strongest influence on nature relatedness changes across all covariates (β = 0.032), although statistical significance again only reached trend level (β = 0.032, p = .058)."
- “Person correlations between predictors and change scores DNR-6BL-2W were significant for two of the five predictor variables…” Which ones? Changed to "Pearson correlations between predictors and change scores ΔNR-6BL-2W were significant for two of the five predictor variables, namely ego-dissolution (EDI) and influence of nature, but not mystical-type (MEQ), challenging (CEQ), or visual experiences (VE)"
- Figure 4. “Nature relatedness changes…”. Relative or absolute? Changed to: "Nature relatedness changes from before to after a psychedelic experience are predicted by acute state and setting" ('Nature relatedness changes' refer to difference scores), which we hoped to make more clear by adding 'from before to after' )
Discussion:
- “…the hypothesis that psychedelics act as biophilia enhancing agents…” Who says that? Does this belong to reference 119 too?
- “It follows that ego-dissolution, which is strongly correlated with the mystical-type experiences occasioned by psychedelics, is linked to the experience of awe,…”
Why “it follows”? removed "it follows that".
- “We found that, at least in the portion of participants with below-average nature relatedness…” But not with the whole dataset? Precisely, this was observed only in the subset of participants with a mean nature relatedness closer to previously assessed samples: "Additionally, we repeated the final GLMM with a restricted sample of participants showing average or below-average nature relatedness at baseline (NR-6BL ≤ 4.01; N = 291). This step was performed to shift the sample closer to a demographically similar reference population of college students (M[NR-6BL ≤ 4.01] = 3.23, vs M = 3.00 in [125] and allow for greater scope of change in the sample, which was otherwise already comparatively high on nature relatedness at baseline – thus, incurring a ceiling-effect and related risk of Type 2 error. "
- “…Studerus et al., 2011 study,…”. Change to “…Studerus and colleagues’ (2011) study…” changed
Conclusion:
- “…assessed in a prospective way.” Conceptually makes most sense for the psychedelic-naïve participants.
General (check the manuscript thoroughly):
- Only a few typos/commas-errors and the like
- Where’s chapter 2.2.? Included now :)
- Layout style: no paragraph indent after a (sub)title Indents changed to follow IJERPH formatting guidelines
References: There are some mistakes, please check again. E. g.: “Nations, U., 2018 revision of world urbanization prospects. 2018, United Nations Department of Economic and Social Affairs. corrected
With kindest regards